# sCD14 Level in Saliva of Children and Adolescents with and without Dental Caries, a Hurdle Model

**DOI:** 10.3390/children8080679

**Published:** 2021-08-04

**Authors:** Gaia Pellegrini, Marcello Maddalone, Matteo Malvezzi, Marilisa Toma, Massimo Del Fabbro, Elena Canciani, Claudia Dellavia

**Affiliations:** 1Department of Biomedical Surgical and Dental Sciences, Università degli Studi di Milano, Via Mangiagalli 31, 20133 Milan, Italy; marilisa.toma@unimi.it (M.T.); massimo.delfabbro@unimi.it (M.D.F.); elena.canciani@unimi.it (E.C.); claudia.dellavia@unimi.it (C.D.); 2Department of Medicine and Surgery, University of Milano-Bicocca, Via Cadore 48, 20900 Monza, Italy; marcello.maddalone@unimib.it; 3Department of Clinical Sciences and Community Health, Università degli Studi di Milano, Via della Commenda 19, 20122 Milan, Italy; matteo.malvezzi@unimi.it; 4IRCCS Galeazzi Orthopedic Institute, Via Riccardo Galeazzi 4, 20161 Milan, Italy

**Keywords:** CD14, saliva, dental caries, children, adolescents, tooth, decay

## Abstract

Objective: Soluble CD14 (sCD14) plays an important role in the innate immune response of the oral cavity. The investigation of this biomarker for detection of carious lesions is an even more actual procedure due to its non-invasiveness and the ease of withdrawal. The purpose of the present observational case-control study was to evaluate whether the quantification of sCD14 in children and adolescent’s saliva can discriminate healthy subjects from those suffering from tooth decay. Materials and Methods: 164 subjects (6 to 17 years) were selected and divided into 2 groups: those with at least 1 decayed tooth were assigned to group Decayed (n = 82) and those free from dental caries to group Healthy (n = 82). The amount of salivary soluble CD14 was quantified. Results: Mean salivary soluble CD14 was 28.3 ± 10.8 μg/mL in the Healthy group and 22 ± 9.6 μg/mL in the Decayed group. A hurdle model was applied to the data to estimate both the probability of having carious lesions and their number in relation to sCD14 levels. sCD14 was strongly associated (*p* < 0.01) with an inverse relation to both the probability of having caries and their number (falling rate of 5% per unit CD14 μg/mL). Conclusions: This data confirms the relationship between sCD14 and the presence of dental caries. However, there is no clear cut off level between healthy and unhealthy subjects, so it is currently not possible to use sCD14 as a biomarker to determine the risk of decays.

## 1. Introduction

The cluster of differentiation 14 (CD14) is the innate immune glicoprotein that interacts with TLRs for pathogens recognition. CD14 is expressed on monocytes/macrophages and neutrophils’ surfaces (mCD14) or secreted in a soluble free form (sCD14). This receptor is able to recognize the lipopolysaccharides (LPS) and peptidoglycans located in Gram + and Gram–bacteria membranes [1]. At the cellular level, mCD14 forms complexes with LPS and peptidoglycans, enhancing recognition by innate immune system cells and pathogenic bacteria phagocytosis and initiates a signal transduction cascade that leads to the production of mediators for the control of infection [2]. The soluble form of this molecule (sCD14) is produced by monocytes via the proteolysis of the membrane-bound receptor, and is then released from the surface or via a protease-independent mechanism to be released from intracellular pools [3]. sCD14 is produced also into the salivary glands and is released by the salivary secretion. sCD14 appears to be capable of mediating the activation by LPS of intestinal epithelial cells that do not present mCD14 through the link with the toll-like receptor-4. This molecule also appears to play an important role in the oral cavity innate immune response by promoting the invasion of oral epithelial cells by Aggregatibacters and consequently increasing the production of interleukin-8 [4,5,6]. The salivary content of sCD14 was found to change as a result of several infectious and inflammatory conditions of the oral cavity including oral lichen planus [7], burning mouth syndrome, periodontal disease [8] and dental caries, and has been proposed as a marker for healthy/diseased status.

Caries in children and adolescents are a global public health [9,10] burden due to their growing prevalence and their disabling consequences. Diagnosis for this disease remains difficult particularly for proximal lesions [11]. For this reason, researchers paid particular attention to the salivary content of this protein as an early and non-invasive method for diagnosis of carious lesions in preschooler and school subjects. However, contrasting data emerged. Some Authors observed a significant rise in salivary sCD14 in patients affected by dental caries [12]. Conversely, other studies indicated higher sCD14 levels in healthy subjects [13,14,15]. These conflicting findings highlight the need for further studies on large sample sizes to define the reliability of sCD14 as a biomarker for the detection of carious lesions.

The investigation of salivary biomarkers to assess oral or systemic diseases/conditions such as SARS-CoV-2 infection is an even more actual procedure due to its non-invasiveness and the ease of withdrawal. Thus, it is particularly adapted for screening of children and subjects with disabilities [16,17].

The purpose of the present study was to evaluate whether in children and adolescents’ salivary levels of sCD14 can discriminate healthy subjects from those suffering from tooth decay.

## 2. Materials and Methods

In this observational case control study, 164 subjects aged 6 to 17 years were selected. For this study patients going at a private dental practice in Milan (Italy) were recruited for programmed periodic dental examinations, or to treat already diagnosed carious lesions.

The study protocol was explained to the patients and their parents/legal guardians, a written IRB-approved Informed Consent Form was provided. The study was performed following the principles outlined in the Declaration of Helsinki (version 2002) on research involving human subjects. All procedures and materials used in the present study were independently reviewed and approved by the ethical committees at the University of Milan (Italy) 18-04-16-12/16. The experiments were undertaken with the understanding and written consent of each subject and according to the above-mentioned principles.

Enrolled subjects were divided into 2 groups: those with at least 1 deciduous or permanent tooth decayed were assigned to group D (Decayed) and those free from dental caries were assigned to group H (Healthy).

### 2.1. Inclusion and Exclusion Criteria

To be included in the study, all subjects were examined clinically by an experienced dentist and fulfilled the following inclusion criteria: age range 6 to 17 years, systemically healthy, no medication intake, no food intolerance, non-smoking and no pathologies of oral soft tissue (i.e., gingivitis, stomatitis). Subjects included in the group H had no decayed, filling or missing teeth due to caries, those included in the group D had at least 1 deciduous or permanent tooth with active carious lesion that was detected and diagnosticated by a visual-tactile method and that needed to be filled. Bitewing radiographs were performed in case of doubt about the presence of interproximal dental caries.

### 2.2. Study Protocol

After inclusion into the study, subjects were visited and their medical history was filled in. In particular, the quality of oral hygiene and the presence of any gastrointestinal disorders/alimentary intolerance were assessed. The visit included the evaluation of the oral health status through the compilation of the index DMFT (decayed, missing, filled, teeth) for the presence of caries.

Before saliva collection, subjects were asked to refrain from eating, drinking or washing teeth for at least 2 h. Saliva was collected from each subject at the beginning of the study, as described previously by Rasperini et al. [16]. Briefly, before to start the study, the subjects removed gross debris by rinsing their mouths with water for about 20 s. After 2 min, spontaneously produced and expectorated saliva was harvested into the plastic funnel placed on a plastic tube. The collection procedure was completed once 2 mL of the saliva was collected or a maximum of 15 min of sampling time was reached. Immediately after collection, samples were placed on ice, supplemented with a proteinase inhibitor and aliquoted before storage at −80 °C until analysis. sCD14 in the saliva samples was quantified by means of a custom human protein array (Quantibody Custom Array; RayBiotech, Inc., Norcross, GA, USA) [18].

Before each assay, whole saliva samples were thawed and microcentrifuged at 500× *g* for 5 min at 5 °C to obtain a cell-free supernatant for analysis. After overnight incubation at 4 °C, samples were washed with wash buffer to remove the unbound material and the detection antibody was added. Wells were incubated and covered with aluminum foil to avoid exposure to light, and fluorescence from each well was detected using a laser scanner (Axon Gene Pix 4300A; MDS Analytical Technologies, Sunnyvale, CA, USA; RayBiotech, Inc., Norcross, GA, USA). The resultant sample signals were compared against the standard curve to determine the sCD14 concentration. Data were extracted and analyzed using the microarray analysis software (RayBio Q Analyzer software; RayBiotech, Inc., Norcross, GA, USA).

### 2.3. Statistical Analysis

The main outcome of the present study was the comparative analysis of the amount of sCD14 in samples of both groups (H0: sCD14 in healthy subjects = sCD14 in decayed patients). Sample size calculation was performed using α = 0.05 (5%) and an 80% sample power. Standard deviation of sCD14 content in saliva of decayed children was 37.67 ng/mL, as obtained in previous papers [19]. The minimum significant value considered was 5.6 ng/mL. On the basis of these data, the needed number of subjects to be enrolled resulted 86 for each group.

Descriptive analysis. In both groups, the mean and standard deviation of clinical parameters and of sCD14 content were computed.

Inferential statistics. To assess if differences in the salivary sCD14 level were present between heathy and decayed subjects, data of two groups (H and D) were compared by *t*-test.

We performed chi-square tests and tests for trends in proportions with an alpha of 0.05 on ordered categorical variables when effects were successively bigger or smaller on sex, age in three developmental age-groups (6–9, 10–13, 14–16 years), oral hygiene, and sCD14 levels in quintiles.

In particular we selected the age groups considering the dentition and pubertal development: 6–9 fist mixed dentition and pre-pubertal stage, 10–13 completion of permanent dentition and developing sexual characteristics, 14–16 permanent dentition and completion of pubertal development [20,21].

We calculated Odds Ratios (OR) and 95% Confidence Intervals (CI) for the binary outcome Healthy vs. Diseased using both univariate and multivariate (adjusting for sex and continuous age) unconditional logistic regression.

After examining the sCD14 distribution between number of carries (0–9) and the results from the logistic models, a hurdle model appeared to be the most appropriate model to examine and describe the data [22]. Briefly, hurdle models are used to evaluate count data with an overabundance of zeros (healthy patients with zero caries in this study). A hurdle model is a 2 steps model that first evaluates a right censored logistic regression to evaluate the probability that an outcome is a zero or not, if this hurdle is passed a left censored count data model is applied to the data to evaluate counts larger or equal to 1. We applied a poisson hurdle model with only sCD14 as a continuous variable since from Vuong tests [23] age, sex and oral hygiene variables, or using a negative binomial distribution in the count part of the model did not give statistically better fits.

The R-project statistical software was used to carry out the analyses [24].

## 3. Results

### 3.1. Study Population

Eighty subjects for the decayed group and eighty-two for the healthy group accepted to participate and were enrolled into the study according to the inclusion/exclusion criteria. The mean age of subjects in the healthy group was 9.9 ± 3.1 and that of patients with caries was 9.2 ± 2.9. Table 1 reports the age, gender distribution and oral hygiene habits of the study population. Six children reported gastric disorders (2 celiac disease, 1 favism) and food intolerance. In the D group the mean number of caries was 3.8 ± 2.1.

### 3.2. sCD14 Levels

After the examination, the salivary collection was conducted by the same operator (MT), the procedure was reported to be easy, rapid, non-invasive and well accepted by all patients. All the harvested saliva samples resulted eligible for the analysis.

The mean salivary soluble concentration of sCD14 was 28.3 ± 10.8 μg/mL (range: 10.3–53.1 μg/mL) in subjects that did not experience dental caries (Healthy-H group) and 22 ± 9.6 μg/mL (range: 3.6–46.6 μg/mL) in subjects with dental caries (Decayed-D group). Data on the salivary protein level separated by age and gender are reported for each group in Table 2.

At the independent *t*-test analysis, sCD14 level resulted significantly higher in healthy than in decayed subjects (*t*-test, *p* < 0.01).

At the within group analyses no differences in the sCD14 amount were found between age groups and between genders. No differences were also found, within each group, in protein amount between subjects that performed oral hygiene procedures once a day and those that performed many times a day.

### 3.3. CD14 and Caries

By inspection we observe an inverse relation between sCD14 and number of caries, of note sCD14 values are distributed along the whole spectrum of values within the healthy group (0 decayed teeth) sCD14 distribution by number of decayed teeth is reported in Figure 1.

Chi-square tests showed that sex, age in developmental groups (6–9, 10–13 and 14–16 years) and oral hygiene were not significantly associated to oral health status in this sample, however the aggregated age variable did have a significative trend (*p* = 0.04), sCD14 was strongly associated (*p* < 0.01) with a significant trend. Unconditional logistic regression (multivariate adjusted for sex and continuous age) resulted in a non significant OR 1.3 (CI 95% 0.6–2.9) for brushing teeth twice a day or more versus once or less. sCD14 in quintiles showed protective ORs compared to the first quintile: 0.17, 0.3, 0.3 and 0.09 for the 2nd, 3rd, 4th and 5th respectively (Table 3).

The hurdle model gives the probability of being in the zero-caries group H according to sCD14 level with 95% CIs (light blue band with red borders) in Figure 2, it successively calculates the expected number of caries if the probability hurdle is passed with 95% CIs (grey band with blue borders). We can see that the probability of being in the H group rises with rising levels of sCD14, while the number of expected lesions drops with rising sCD14.

The model allowed to calculate the probability of belonging to the D group with bootstrapped 95% CIs according to continuous sCD14 levels as summarized in Table 4 for selected sCD14 levels in μg/mL.

Table 5 gives the expected numbers of decayed teeth with 95% CIs for the same selected values of sCD14 in μg/mL in the case where the probability hurdle is passed.

## 4. Discussion

The assessment of the presence of dental caries by means of a non-invasive and ease of withdrawal method is even more clinically important since it may reduce the number of bitewing radiographs for their detection and facilitate their early diagnosis in particular in patients difficult to visit or with disabilities.

The main purpose of the present study was to evaluate whether the salivary content of soluble CD14 was different between subjects with active dental caries or without carious lesions. At the immune assay analysis, saliva sCD14 content resulted significantly lower in patients with decayed teeth than in healthy subjects. This data seems to confirm the relationship between the investigated protein and the development of dental caries and are in accordance with the literature [12,13,14,15]. Authors detected the experimental protein by ELISA test and Western blot and reported lower levels, or even the complete absence [15], of sCD14 in patients with at least one carious lesion [13,14,15] than in caries free subjects. These findings together with results of the present study confirm an inverse correlation between the protein and the number of decayed teeth.

To evaluate whether the level of sCD14 was indicative of the presence/absence of dental caries or rather it was indicative of the numerosity of these lesions, first sCD14 levels were analyzed in quintiles to investigate whether there was a dose response gradient, Chi-squared tests and portional trend tests confirmed this. We further quantified this relation with a multivariate unconditional logistic regression that showed successively stronger inverse associations with higher sCD14 quintiles. However, in order to describe both the logistic like binary behavior for the presence or lack thereof of caries, and the count like nature of the number of caries per patient in relation to sCD14 levels a 2 step hurdle model was required [22]. The hurdle model allows to model and estimate both the effect of sCD14 levels on the probability of having carious lesions vs. being in the healthy group, and successively if the probability of being healthy is surpassed (the hurdle is jumped), it allows to estimate the number of these lesions in relation to sCD14 levels. The estimates from this model confirm the results from the logistic regression, i.e., those patients with higher levels of sCD14 are more likely to be caries free, but also model the expected number of lesions in diseased patients according to sCD14 levels. These findings are interesting and noteworthy as of themselves and the hurdle model shows a good fit in describing the sCD14 caries relation, however the answer to the clinical question of whether sCD14 makes a good marker for a screening test for dental health is mostly negative. The probability function of caries presence with levels of sCD14 shows that there is no clear cut off level between healthy and unhealthy subjects, and the values that allow for a clearer discrimination between healthy and diseased subjects are extreme values at both ends of the spectrum with very few subjects, and hence of scarce practical use. Therefore, sCD14 on its own is not a reliable biomarker for detection of dental caries.

The present as well as most of the previous studies were conducted in a population of children/adolescents with ages of 2–6 [13,14] and 6–12 years [15]. The overall incidence of caries in deciduous and permanent teeth in preschool and school children is still high and involves about half the population [25]. This public health problem is aggravated by neglect since, in 2010, untreated caries in deciduous teeth was the 10th-most prevalent condition and affected about 9% of the global population with a significant impact on their oral-health related quality of life [26]. Considering permanent teeth, this prevalence reached 35% and represented the most prevalent condition of the global population in 2010 [27]. In children, one important factor affecting the incidence of dental caries is the fear of dentists that significantly reduces the possibilities for a specialist to carry out an adequate diagnostic, therapeutic and preventive program [28]. The early diagnosis of dental caries is fundamental for a conservative management of this disease and for prevention of tooth extraction. Several non-invasive methods can be used for the detection and diagnosis of dental caries such as visual-tactile, Quantitative Light-induced Fluorescence, DIAGNOdent, Fibre-optic Transillumination and Electrical Conductance [29]. However, all these methods are chair-side. The development of an easy, non-invasive and non-chair-side test for the early detection of dental caries may help to overcome the fear and allow diagnosis before the disease becomes irreparable and the teeth need extraction. Tooth loss has detrimental effects on many aspects of children’s lives, such as occlusion [30], speech [31] and language development, masticatory function, esthetics, oral health-related quality of life [32]. For this reason, the present study as well as most of the previous studies on this topic were designed on a young population.

Analyzing the data, no differences in sCD14 levels resulted between age group and gender. This seems to indicate that the salivary content of this protein is not affected by hormonal factors or by the development status of the immune system.

From data of the present study, the tested protein cannot be used as marker for active caries on the overall children and adolescent population. Infact, sCD14 amount resulted not indicative of the presence/absence of dental caries. Furthermore, even if an inverse correlation was found between the protein level and the number of decayed teeth in accordance to previous studies [12,13,14,15], the high interindividual variability that was found also in literature, does not allow to establish a range standard for all subjects of correspondence between sCD14 amount and degree of this dental disease. However, it may be possible to combine it with other markers/lifestyle risk factors that are known to be associated with carious lesions and construct a multivariate probability profile that is more capable of discriminating between healthy and diseased subjects. Within the limitations of the current study is that the inflammatory gingival status of subjects has not been investigated. Since the periodontal disease may affect salivary levels of sCD14, further studies may be designed considering this condition as a factor modifying the protein amount [33].

A previous study investigated the effect of a restorative intervention on the concentration of sCD14 in saliva [15]. In the present study, this second analysis has not been performed since the aim was not to detect the changes in the protein concentration but to represent the status of sCD14 salivary content at the time of the visit and of the detection of dental caries.

## 5. Conclusions

To conclude, data of the present study confirm the association between salivary levels of sCD14 and amount of dental caries. However, since this biomarker did not result indicative of the presence/absence of dental disease, the analysis of salivary sCD14 levels on their own as a non-invasive method to screen the risk of caries in preschooler and schooler population is not reliable. The importance to find a non-invasive and easy methods for assessment of oral diseases in children may bring to combine sCD14 with other factors into a multivariate indicator for the creation of a clinical protocol of screening.

## Figures and Tables

**Figure 1 children-08-00679-f001:**
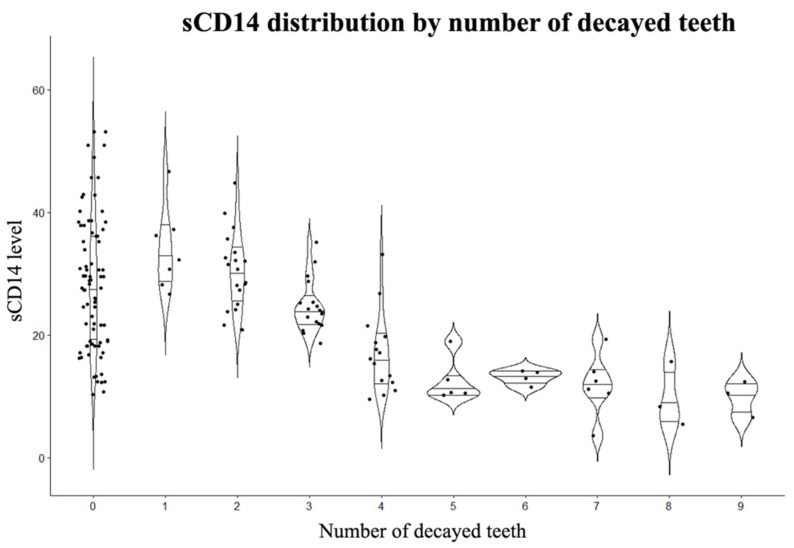
sCD14 distribution by number of decayed teeth violin plot showing density distribution with quartiles of sCD14 for each number of decayed teeth.

**Figure 2 children-08-00679-f002:**
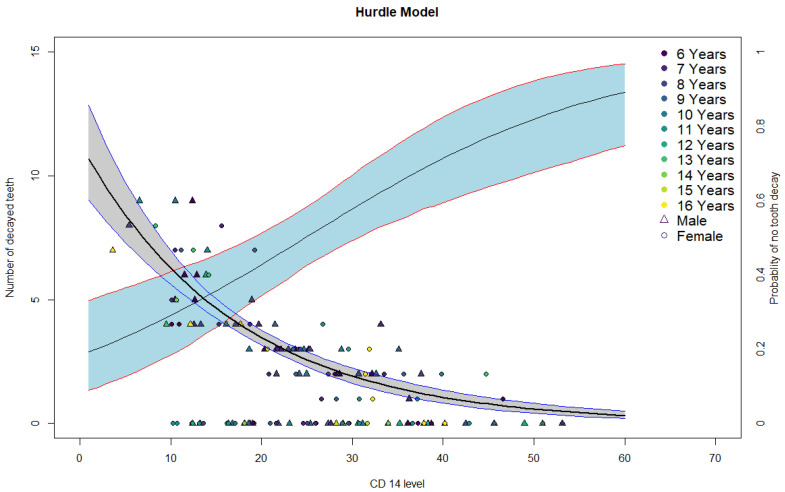
Plot of hurdle model results, triangles are male subjects and females are circles, subjects are colored by age. The light blue band with red borders is the 95% CI of the probability estimate of being in the H group, the grey band is the 95% CI of the estimate of expected number of lesions in relation to sCD14 levels.

**Table 1 children-08-00679-t001:** Study population distribution with respect to oral health status according to sex, age in three groups, oral hygiene, and sCD14 in quintiles, with Chi-square, and Chi-square test for proportional trends *p*-values.

		Healthy N (%)	Decayed N (%)	X^2^ P	X^2^ P Trend
**Gender**	Male	36 (46.2%)	42 (53.8%)	0.348	0.274
	Female	46 (54.8%)	38 (45.2%)		
**Age in tertiles**	6–9 years	39 (42.9%)	52 (57.7%)	0.081	0.037
	10–13 years	30 (60.0%)	20 (40.0%)		
	14–16 years	13 (61.9%)	8 (38.1%)		
**Oral Hygiene**	Brushes 2 or more times daily	66 (52.0%)	61 (48.0%)	0.642	0.512
	Brushes less than twice daily	16 (45.7%)	19 (54.3%)		
**CD14 μ** **g/mL in quintiles**	Less than 15.30	8 (24.2%)	25 (75.8%)	0.001	0.001
	15.31–21.61	19 (59.4%)	13 (40.6%)		
	21.62–27.63	14 (43.8%)	18 (56.2%)		
	27.64–33.92	16 (50.0%)	16 (50.0%)		
	33.91–53.11	24 (75.0%)	8 (25.0%)		

**Table 2 children-08-00679-t002:** Data on sCD14 salivary levels (μg/mL) reported according to gender and age.

	Healthy	Decayed
**Gender**		
Girls	27.1 ± 11.0	24.1 ± 10.1
Boys	29.7 ± 10.7	20.1 ± 8.7
**Age group (years)**		
6–9	27.6 ± 9.4	22.9 ± 8.9
10–13	28.3 ± 12.7	22.5 ± 10.9
14–16	30.3 ± 11	18.7 ± 10.9

**Table 3 children-08-00679-t003:** Unconditional logistic regression univariate and multivariate (adjusted for sex and age) ORs with 95% CIs.

	OR	95% CI	Multivariate OR	95% CI
**Oral_Hygiene:**				
Brushing once or less vs. more times a day	1.285	0.607–2.748	1.286	0.575–2.917
**sCD14>**				
2nd quintile vs. 1st	0.219	0.072–0.616	0.172	0.049–0.541
3rd quintile vs. 1st	0.411	0.138–1.165	0.305	0.088–0.968
4th quintile vs. 1st	0.320	0.107–0.898	0.301	0.087–0.960
5th quintile vs. 1st	0.107	0.032–0.315	0.086	0.023–0.278

**Table 4 children-08-00679-t004:** Probability of belonging to group D with 95% CI for selected values of sCD14 in μg/mL.

sCD14 μg/mL	Probability Estimate	95% CI
1	0.81	0.67–0.91
10	0.71	0.59–0.82
20	0.57	0.49–0.66
30	0.42	0.33–0.51
40	0.29	0.18–0.41
50	0.18	0.08–0.32
60	0.11	0.03–0.25

**Table 5 children-08-00679-t005:** Expected numbers of decayed teeth with 95% CIs for the same selected values of sCD14 in μg/mL in the case where the probability hurdle is passed.

sCD 14 μg/mL	Number of Expected Caries	95% CI
1	10.69	9.02–12.86
10	6.27	5.61–6.98
20	3.47	3.16–3.78
30	1.92	1.63–2.25
40	1.06	0.82–1.35
50	0.59	0.41–0.82
60	0.32	0.2–0.5

## Data Availability

The data that support the findings of this study are available on request from the corresponding author, [GP]. The data are not publicly available due to privacy restriction.

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
