# Peer review of "sCD14 Level in Saliva of Children and Adolescents with and without Dental Caries, a Hurdle Model"

_children, 2021, doi:10.3390/children8080679_

Round 1

Reviewer 1 Report

I have a question for authors: 

Why did you use just a simple and so common method in order to detect and diagnose the active carious lesions in subjects? (visual-tactile method).

Thank you.

Author Response

I have a question for authors: 

Why did you use just a simple and so common method in order to detect and diagnose the active carious lesions in subjects? (visual-tactile method).

Answer: thank you for the question. During the activities of screening, the use of simple methods is preferred. However, in the present study, bitewing radiographs were performed in case of doubt about the presence of interproximal dental caries to have accurate data. The following sentence has been added in the materials and methods: “Bitewing radiographs were performed in case of doubt about the presence of interproximal dental caries.”

Changes within the text have been highlighted in green.

Reviewer 2 Report

Manuscript has been prepared properly. Well-described methodology of the research is followed by the adequate discussion over the results of the analyses. Minor changes are needed.

  • Abstract of the manuscript is supplemented with the brief information concerning the novelty of the study.
  • Additional paragraph concerning the mechanism of the sCD14 formation needs to be added to the Introduction section (some sentences concerning this issue have been written but this subject should be more widely described). 
  • Please, characterize briefly the reliability of sCD14 as a biomarker for the detection of carious lesions. 
  • The importance and clinical relevance of sCD14 should be discussed in more detail by Authors.
  • In line 273 - "The tested protein cannot be used as a marker for active caries" -explain why and compare with other studies.
  • Limitations of the study should be added.
  • In Conclusion - the highlights of the research should be clearly indicated and clinical relevance. 

Author Response

Dear reviewer,

thank you for your kind revision. We modified the manuscript following your suggestions and we think that it remarkably improved. Changes have been highlighted in yellow.

Reviewer 2:

Manuscript has been prepared properly. Well-described methodology of the research is followed by the adequate discussion over the results of the analyses. Minor changes are needed.

  • Abstract of the manuscript is supplemented with the brief information concerning the novelty of the study.

Answer: thank you. The following sentence has been added in the abstract: “The investigation of this biomarkers for detection of carious lesions is an even more actual procedure due to its non-invasiveness and the ease of withdrawal.”

  • Additional paragraph concerning the mechanism of the sCD14 formation needs to be added to the Introduction section (some sentences concerning this issue have been written but this subject should be more widely described). 

Answer: thank you. The first part of the introduction has been modified as follows:

The cluster of differentiation 14 (CD14) is innate immune glicoprotein that interacts with TLRs for recognition of pathogens. CD14 is expressed on monocytes/macrophages and neutrophils’ surfaces (mCD14) or secreted in a soluble free form (sCD14). This receptor is able to recognize the lipopolysaccharides (LPS) and peptidoglycans located in Gram + and Gram – bacteria membranes [1]. At the cellular level, mCD14 forms complexes with LPS and peptidoglycans, enhancing recognition by innate immune system cells and pathogenic bacteria phagocytosis and initiates a signal transduction cascade that leads to the production of mediators for the control of infection [2]. This molecule’s soluble form (sCD14) is produced by monocytes either through the proteolysis of the membrane-bound receptor then is released from the surface or by a protease-independent mechanism then is released from intracellular pools [3]. sCD14 is produced also into the salivary glands and is released by the salivary secretion

  • Please, characterize briefly the reliability of sCD14 as a biomarker for the detection of carious lesions. 

Answer: the following sentence has been added in the discussion: “Therefore, sCD14 on its own is not a reliable biomarker for detection of dental caries.”

  • The importance and clinical relevance of sCD14 should be discussed in more detail by Authors.

Answer: the following sentence has been added in the discussion: “The assessment of the presence of dental caries by means of a non-invasive and ease of withdrawal method is even more clinically important since it may reduce the number of bitewing radiographs for their detection and facilitate their early diagnosis in particular in patients difficult to visit or with disabilities.”

  • In line 273 - "The tested protein cannot be used as a marker for active caries" -explain why and compare with other studies.

Answer: The following sentence was added: “Infact, sCD14 resulted not indicative of the presence/absence of dental caries. Furthermore, even if an inverse correlation was found between the protein level and the number of decayed teeth in accordance to previous studies [12-15], the high interindividual variability that was found also in literature, does not allow to establish a range standard for all subjects of correspondence between sCD14 amount and degree of this dental disease.”

  • Limitations of the study should be added.

Answer: the present limitation has been added in the discussion:“ Within the limitations of the current study is that the inflammatory gingival status of subjects has not been investigated. Since the periodontal disease may affect salivary levels of sCD14, further studies may be designed considering this condition as a factor modifying the protein amount [33]”

  • In Conclusion - the highlights of the research should be clearly indicated and clinical relevance. 

Answer: the conclusion has been rewritten: “To conclude, data of the present study confirm the association between salivary levels of sCD14 and amount of dental caries. However, since this biomarker did not resulted indicative of the presence/absence of dental disease, the analysis of salivary sCD14 levels on their own as a non-invasive method to screen the risk of caries in preschooler and schooler population is not reliable. The importance to find a non-invasive and easy methods for assessment of oral diseases in children may bring to combine sCD14 with other factors into a multivariate indicator for the creation of a clinical protocol of screening.”
